# An Adjuvant Stem Cell Patch with Coronary Artery Bypass Graft Surgery Improves Diastolic Recovery in Porcine Hibernating Myocardium

**DOI:** 10.3390/ijms24065475

**Published:** 2023-03-13

**Authors:** Rishav Aggarwal, Koray N. Potel, Annie Shao, Simon W. So, Cory Swingen, Christina P. Reyes, Rebecca Rose, Christin Wright, Laura L. Hocum Stone, Edward O. McFalls, Tammy A. Butterick, Rosemary F. Kelly

**Affiliations:** 1Division of Cardiothoracic Surgery, Department of Surgery, University of Minnesota Medical School, Minneapolis, MN 55455, USA; aggar122@umn.edu (R.A.); shaox091@umn.edu (A.S.); swing001@umn.edu (C.S.); reyes147@umn.edu (C.P.R.); rose0447@umn.edu (R.R.); wrig0510@umn.edu (C.W.); stone337@umn.edu (L.L.H.S.); 2School of Medicine, Dentistry and Biomedical Sciences, Queen’s University Belfast, Belfast BT9 7BL, UK; kpotel01@qub.ac.uk; 3Department of Neuroscience, University of Minnesota, Minneapolis, MN 55455, USA; soxxx063@umn.edu (S.W.S.); butte017@umn.edu (T.A.B.); 4Department of Research, Center for Veterans Research and Education, Minneapolis, MN 55417, USA; 5Division of Cardiology, Richmond VA Medical Center, Richmond, VA 23249, USA; mcfal001@umn.edu

**Keywords:** hibernating myocardium, diastolic dysfunction, mesenchymal stem cell, ischemic heart disease, cardiac MRI, CABG, PGC1α

## Abstract

Diastolic dysfunction persists despite coronary artery bypass graft surgery (CABG) in patients with hibernating myocardium (HIB). We studied whether the adjunctive use of a mesenchymal stem cells (MSCs) patch during CABG improves diastolic function by reducing inflammation and fibrosis. HIB was induced in juvenile swine by placing a constrictor on the left anterior descending (LAD) artery, causing myocardial ischemia without infarction. At 12 weeks, CABG was performed using the left-internal-mammary-artery (LIMA)-to-LAD graft with or without placement of an epicardial vicryl patch embedded with MSCs, followed by four weeks of recovery. The animals underwent cardiac magnetic resonance imaging (MRI) prior to sacrifice, and tissue from septal and LAD regions were collected to assess for fibrosis and analyze mitochondrial and nuclear isolates. During low-dose dobutamine infusion, diastolic function was significantly reduced in HIB compared to the control, with significant improvement after CABG + MSC treatment. In HIB, we observed increased inflammation and fibrosis without transmural scarring, along with decreased peroxisome proliferator-activated receptor-gamma coactivator (PGC1α), which could be a possible mechanism underlying diastolic dysfunction. Improvement in PGC1α and diastolic function was noted with revascularization and MSCs, along with decreased inflammatory signaling and fibrosis. These findings suggest that adjuvant cell-based therapy during CABG may recover diastolic function by reducing oxidant stress–inflammatory signaling and myofibroblast presence in the myocardial tissue.

## 1. Introduction

In patients with ischemic heart disease, coronary artery bypass graft (CABG) surgically restores the blood flow and oxygen supply to improve long-term survival but does not prevent the risk of congestive heart failure [1,2]. At rest, hibernating myocardium (HIB) is a disease state with reduced blood supply and function while maintaining viability and variable contractile reserve [3]. Without treatment, HIB may progress to heart failure as cardiac function becomes increasingly depressed, especially under chronic ischemic events or during the increased workload [4]. The optimal therapy for HIB is timely, complete revascularization with CABG to restore blood flow and avoid progression to heart failure. However, although the revascularization of HIB should restore contractile function, clinical studies and our animal model demonstrate that recovery is often incomplete [5]. Importantly, diastolic dysfunction, a predictor of adverse outcomes following revascularization, may also persist after CABG, suggesting the need for adjunctive therapeutics, potentially at the time of CABG [6]. Currently, there is no clinically available adjunctive therapy to be used with CABG to restore myocytes to full functional capacity. This is a major therapeutic gap given the large number of patients that progress to heart failure despite appropriate revascularization [7].

Our group has a well-established porcine model of HIB that is amenable to revascularization, providing an animal model that simulates chronic ischemia [8]. This model of HIB has allowed us to characterize the impairment in systolic function, measured by regional wall thickening (measured at the end diastole and end systole) and circumferential strain, that improves with revascularization [4]. Our model has shown impaired diastolic function, measured by the Peak Filling Rate (PFR), that only slightly improves with revascularization alone [9]. The proteomic profile of HIB is characterized by an upregulation of oxidative stress and inflammatory cytokines as well as the downregulation of mitochondrial and contractile proteins [10,11]. Our molecular analyses have exhibited that the mitochondrial proteins involved in oxidative metabolism remain persistently reduced despite revascularization, which could be a possible cause of the incomplete functional recovery [12,13].

To address this gap in the treatment of HIB, additional therapies are needed to fully restore the cardiac function to pre-ischemic levels by targeting the remaining metabolic deficits in revascularized myocardium. Recent studies have uncovered the important role that the peroxisome proliferator-activated receptor-gamma coactivator (PGC1α) plays in mitochondrial metabolism as well as its regulatory effects on oxidative stress and inflammation [14,15]. HIB presents a unique target for stem cell therapy as it is fully viable but has dysfunctional myocardium with the potential to be rescued. Mesenchymal stem cells (MSCs) are an ideal candidate for treatment, as previous studies have shown they have the potential to increase PGC1α, which is a key mediator of mitochondrial biogenesis [16]. PGC1α might therefore be a promising target of MSCs for overcoming the current limitations in the treatment of chronically ischemic myocardium.

Adult-derived stem cells are promising adjunctive therapeutics, as they are readily accessible and can expand ex vivo to allow for symmetric stem cell division and self-renewal without further differentiation [17,18]. Considering the potential of MSCs to enhance PGC1α signaling [19], this study focuses on highlighting the therapeutic potential of an MSC cardiac patch that can be used at the time of CABG to fully reverse the hibernation phenotype and restore optimal cellular metabolism, resulting in the full recovery of diastolic function in chronically ischemic myocardium.

## 2. Results

### 2.1. Physiologic Analysis

Cardiac MRI showed preserved left ventricular ejection fraction (LVEF) among all four groups (Control, HIB, CABG, and CABG + MSC) at rest (56 ± 7.2%, 55 ± 10.7%, 57.5 ± 7.5%, and 54.7 ± 8.1%, respectively) and a significant increase in LVEF under stress with low-dose dobutamine infusion at 5 µg/kg/min (74.2 ± 11.2%, 65.7 ± 11.1%, 70 ± 8.1%, and 69.8 ± 8.6% respectively; *p* < 0.01 for all groups). The ΔLVEF (LVEF (dobutamine)–LVEF (rest)) was also measured for all four groups (18.2%, 10.7%, 12.5%, and 15.1%, respectively) and showed significant differences in the increase in LVEF (*p* < 0.05).

Under stress, using the low-dose dobutamine infusion at 5 µg/kg/min, there was a significant increase in the Peak filling rate/End diastolic volume (PFR/EDV) in all four groups (6.88 ± 1.43, 5.49 ± 0.77, 6.25 ± 0.87, and 6.62 ± 1.06, respectively; *p* < 0.01 for all groups), with no significant differences in ΔPFR/EDV (PFR/EDV (dobutamine)–PFR/EDV (rest)) (2.45, 1.23, 1.56, and 1.71, respectively). The mean percentage change in PFR/EDV between rest and dobutamine among the four groups does not show a significant difference (56.4%, 31.5%, 36.7%, and 38.9%, respectively). However, there was a statistically significant decrease in PFR/EDV in the HIB compared to the control group (5.49 ± 0.77 vs. 6.88 ± 1.43, respectively; *p* < 0.05). In CABG, there was a trend towards improvement (not statistically significant) in PFR/EDV compared to the HIB group (6.25 ± 0.87 vs. 5.49 ± 0.77, respectively; *p* = 0.06). However, in CABG + MSC, there was a statistically significant increase in PFR/EDV when compared to the HIB group (6.62 ± 1.06 vs. 5.49 ± 0.77, respectively; *p* = 0.03) (Figure 1). Similar to our previous work, the absence of necrosis as well as the patency of the left-internal-mammary-artery (LIMA)-to-the-left-anterior-descending-artery (LAD) bypass graft distal to the area of stenosis was confirmed by using cardiac MRI [13].

### 2.2. Histologic Analysis

Trichrome staining was used to demonstrate collagen infiltration into different layers of connective tissue in three experimental groups compared to the control (Figure 2A–D). Both fibrosis and inflammation were graded or scored using qualitative analysis, as seen in Table 1. The animals in the control group had a score of 1A, the HIB animals had scores of 4D or 5D, the CABG animals showed scores of 2B or 3C, and the CABG + MSC group showed minimal fibrosis and inflammation, as seen in the control group, with a score of 1A. The presence of myofibroblasts was assessed using alpha smooth muscle actin (SMA) staining. In the control group, there was an absence of alpha-SMA stain in the myocardial tissue beyond the expected smooth muscle in the microvasculature. The HIB group showed increased myofibroblasts in areas of fibrosis. The CABG group showed very few myofibroblasts associated with alpha-SMA stain, and there were no myofibroblasts seen in the CABG + MSC group (Figure 2E–H).

### 2.3. Inflammatory Signals

The transcriptional coactivator Peroxisome Proliferator-Activator Receptor Gamma Coactivator 1-Alpha (PGC1α) has been shown to be a key mediator in mitochondrial-related alterations, and it has regulatory effects on oxidative stress and inflammation in chronically ischemic hearts and revascularization [22,23]. An analysis of western blots of CABG + MSC myocardial tissue shows a significant increase in the PGC1α signal (0.09 ± 0.01) compared to the HIB (0.01 ± 0.01, *p* < 0.001) and CABG groups (0.05 ± 0.01, *p* = 0.02) (Figure 3A). The expression of Nuclear Factor Kappa-Light-Chain Enhancer of Activated B Cells (NFκB), a proinflammatory marker, showed a dramatic decrease in CABG + MSC (0.73 ± 0.10) compared to the HIB (1.53 ± 0.11, *p* < 0.0001) and CABG groups (1.05 ± 0.10, *p* = 0.01) (Figure 3B). We focused on analyzing only the experimental groups (HIB, CABG, and CABG+MSC) for western blots. These groups underwent hibernation surgery, followed by revascularization and the placement of an epicardial MSC patch. Animals that underwent thoracotomy and had their pericardium removed experience a very different stress response and inflammatory changes associated with surgery compared to the control group.

A multiplex cytokine and chemokine array analysis was performed to measure the inflammasome of the four study groups. Granulocyte-macrophage colony stimulating factor (GM-CSF) expression showed a dramatic decrease in CABG + MSC when compared to the HIB group (0.5302 ± 0.04 vs. 2.581 ± 0.31, respectively; *p* < 0.0001). Interferon gamma (IFNγ) expression showed a statistically significant decrease in CABG + MSC compared to the HIB group (1.411 ± 0.22 vs. 3.178 ± 0.23, respectively; *p* < 0.0001). Similarly, IL-2 and IL-1β were found to be significantly lower in CABG + MSC compared to the HIB group (0.017 ± 0.04 vs. 0.316 ± 0.04, *p* < 0.0001, and 0.006 ± 0.02 vs. 0.191 ± 0.03, *p* < 0.0001, respectively) (Figure 4A–D).

## 3. Discussion

In this study, we found that treatment with an allogeneic MSC-loaded patch during CABG dramatically improves diastolic function upon inotropic stimulation. We have previously presented our swine model of HIB, which recapitulates the clinical observations initially described by Rahimtoola et al., and found that diastolic function in HIB remains impaired despite revascularization with CABG [3,9]. While surgical revascularization with CABG can restore blood flow to previously ischemic myocardium and reduces long-term mortality, its role in preventing subsequent progression to diastolic heart failure is unknown [1,2]. The remaining functional impairment might be due to persistent metabolic adaptations, which do not fully recover after the sufficient restoration of blood supply and prevent a complete return to the preischemic state.

Hibernation involves a complex program of cytoprotective gene expression changes, which minimize cell injury and tissue necrosis by downregulating oxidative metabolism, upregulating glucose uptake, and limiting contractile activity [10,24]. These protective adaptations aim to reestablish the equilibrium between oxygen supply and metabolic demands, which is disrupted due to coronary artery disease and preserves tissue viability at the expense of cardiac function [3,11].

The downregulation of the transcriptional coactivator PGC1α, an important regulator of mitochondrial metabolism, oxidative stress, and inflammation, facilitates this metabolic shift [25]. PGC1α signaling enhances mitochondrial biogenesis while reducing oxidative stress and inflammation by inducing antioxidant enzymes and inhibiting NFκB [15]. Oxidative stress and inflammatory signaling are known to be increased in response to myocardial ischemia and are observed in HIB. Although proinflammatory signaling might have cytoprotective functions in acute ischemia by regulating cell death and contractility, it is known to contribute to tissue injury and fibrosis in chronic ischemia [26]. A chronic persistence of the metabolic and functional adaptations seen in HIB, even after the restoration of sufficient blood supply through CABG, might explain the remaining impairment in contractile function we observed in our swine model. Hence, reversing these hibernation features through adjunctive therapies used alongside CABG, such as stem cell-based agents, might lead to the full recovery of contractile function.

Their innate ability to home in on sites of injury as well as their low immunogenicity are reasons why bone marrow-derived stem cells have been studied as potential cell-therapeutic agents [27,28,29,30]. However, stem cells are unable to revitalize already infarcted tissue, which is why they previously showed a limited potential when applied to models of myocardial infarction. Nevertheless, studies were able to show benefits of stem cell therapy in peri-infarct myocardial regions [31]. HIB, much like peri-infarcted myocardium, is able to recover its function, which is why MSC treatment has a therapeutic potential in this chronically ischemic but still viable tissue.

Despite a wealth of research, the exact mechanisms by which stem cells improve outcomes in cardiac diseases remain poorly understood. Previous methods of administration including intracoronary injection and direct myocardial injection have resulted in poor cell retention in the targeted myocardium [32,33].

MSCs embedded onto absorbable vicryl patches have shown greater cell retention and improved cardiac function [34]. This suggests that MSCs might act on injured myocardium through unknown paracrine reparative actions, rather than simply replacing injured myocytes.

In a previous study, we observed no improvement in systolic functional recovery following adjunctive therapies such as CoQ10 [35]. When using an adjunctive MSC patch, however, we saw an improvement in systolic function without a significant increase in regional blood flow. This suggests that the underlying regenerative mechanism may be independent of coronary blood flow [16]. In this study, adjunctive MSC therapy during CABG restored the parameters of diastolic filling and increased the expression of PGC1α, which supports the findings of other investigators [36]. We were able to show that the therapeutic effect of our MSC patch is not due to structural support from the patch itself [16] but likely to the consequence of alterations in cellular metabolism, as well as oxidative and inflammatory signaling [16]. These finding support the notion that stem cell-based therapies might be able to reverse the persisting hibernation-induced changes following CABG, thereby improving the myocardial functional recovery beyond surgical revascularization.

Deng et al. recently showed that mitochondria are key targets when trying to reduce fibrosis in a murine model of heart failure with preserved ejection fraction (HFpEF) [37]. HIB is characterized by the downregulation of mitochondrial electron transport chain protein expression, which persists despite CABG [38]. Treatment with an MSC patch has shown dramatic increases in the expression of electron transport chain components, including ATP synthase and succinate dehydrogenase, which are critical to mitochondrial ATP production [16]. In this study, we have shown a dramatic increase in the expression of PGC1α with the addition of MSCs compared to HIB or CABG alone. These observations suggest that MSCs improve cardiac function through mitochondria-based mechanisms. The prooxidant and proinflammatory signaling pathways that promote tissue fibrosis and contractile dysfunction may be counterbalanced by the inhibitory effects of enhanced PGC1α signaling (Figure 5).

Oxidative stress and inflammation secondary to chronic ischemia can cause pathological remodeling of the heart [39]. The associated conversion of fibroblasts to myofibroblasts increases collagen formation, particularly in septal regions. Prolonged or repeated ischemia activates the transcription factor NFκB, which upregulates pro-inflammatory cytokines including GM-CSF, IFN γ, IL-2, and IL-1β. These signaling agents facilitate the crosstalk between myocytes and infiltrating fibroblasts, while modifying energy metabolism via PGC1α downregulation [39]. In this study, we were able to show enhanced NFκB-mediated inflammatory signaling in HIB, which significantly decreased with the addition of MSCs compared to HIB or CABG alone. This supports the notion that MSCs provide regenerative benefits beyond the mere reestablishment of blood flow.

In our animal model of hibernating myocardium, adjuvant stem cell therapy during revascularization decreases pro-inflammatory cytokine expression and increases mitochondrial biogenesis, resulting in a significant improvement in diastolic function. This therapy may further improve outcomes for patients suffering from ischemic heart disease by fully reversing the phenotypic adaptations of myocardial hibernation and restoring optimal cellular metabolism. Our results therefore highlight the potential of using stem cell-based therapies, such as an MSC-loaded patch, in order to address the limitations of current approaches, including CABG, and potentially enable the full functional recovery of chronically ischemic myocardium. Future steps towards clinical utilization must focus on the development of an “off-the-shelf” patch, which has been proven to be safe in humans and can be widely applied as an adjunctive therapy for ischemic heart disease.

### Limitations

Our study has several limitations. Although adult swine might be a clinically more appropriate model, juvenile animals were used in this study due to the size limitations of surgical and imaging equipment. Furthermore, this model of single-vessel stenosis without comorbidities does not fully reflect the extent and effects of long-standing atherosclerosis. Future directions will focus on studying multivessel disease in a porcine model of HIB. Additionally, although a second operation adds to the complexity of the model, we performed CABG rather than percutaneous cutaneous intervention due to the rigid nature of the constrictor that was placed. Although the four-week recovery period post-CABG limits our ability to assess acute as well as late inflammatory and mitochondrial responses, we did not observe further improvement in previous studies with up to three months of recovery [13]. It was necessary to exclusively use female swine in order to allow for the socialization of animals during surgical recovery time, as required by the Institutional Animal Care and Use Committees, and to ensure the safety of personnel by limiting the handling of large, tusked boars. We only used low-dose dobutamine infusion to induce stress. Higher doses of dobutamine would cause tachyarrhythmias and limit the reliability of the measurements of diastolic function. We did not seek to use treadmill exercise, as it requires more laboratory personnel and animal training to complete the studies, and the test is not reliable and not as reproducible as a test in pigs. Additionally, the lower number of animals in the “CABG+MSC” group might be considered a limitation, which we aim to address by including more animals in our future analyses.

Our treatment groups were not randomly assigned. However, our collaborators, including radiologists and pathologists, were blinded. It can be assumed that the hearts from this group were within normal parameters prior to hibernation, for which pre-treatment MRIs were not performed. Following our observation of diastolic dysfunction, as described above, we aim to use invasive hemodynamic measurements, including the end-diastolic pressure and relaxation time constant, in future studies.

Lastly, although we would like to determine the fate of MSCs following sacrifice, the time from patch application to termination is too long to allow for the utilization of current cell-labeling technologies.

## 4. Materials and Methods

### 4.1. Animal Use

This study was approved by the Institutional Animal Care and Use Committees of the Minneapolis VA Medical Center and the University of Minnesota and was performed conforming to the National Institutes of Health guidelines. A priori power analyses using G * Power software (Heinrich Heine University, Dusseldorf, Germany) were used to determine group sizes.

### 4.2. Study Design

We have previously characterized our swine model of HIB [40]. Please refer to Aggarwal et al. for detailed methods [9]. Briefly, 51 juvenile female Yorkshire-Landrace swine were studied with the following groups: Control (n = 6), HIB (n = 17), CABG (n = 19), and CABG + MSC (n = 9). The groups were matched for sex, age, diet, and weight. Of 51 animals, 45 were subjected to a hibernation procedure, during which a constrictor was placed on the left anterior descending artery (LAD) to cause chronic ischemia without infarction and induce the HIB phenotype over the following 12 weeks. At the end of 12 weeks and prior to revascularization, the first cardiac MRI was obtained to assess for LAD stenosis and systolic and diastolic function in hibernating animals. Subsequently, animals were assigned to termination or revascularization with or without an MSC patch. CABG was performed using the LIMA-to-LAD graft, and an epicardial patch with MSCs was applied to the hibernating myocardial region in the CABG + MSC group animals. The CABG and CABG+MSC groups were allowed to recover for four weeks. Prior to termination, a second cardiac MRI was performed to assess global and regional function and coronary anatomy and to rule out infarction by using gadolinium contrast. After termination, myocardial tissue was obtained for histologic and molecular analyses. Our previous studies using sham operations for control animals showed no physiologic or metabolic difference from normal, non-operated animals. Hence, the controls in this study did not undergo sham operations [16,38].

### 4.3. Off-Pump CABG + Mesenchymal Stem Cell (MSC) Patch Placement

The swine model of off-pump CABG has also been previously well characterized [13]. Refer to Aggarwal et al. for detailed methods [9]. Briefly, at the end of 12 weeks after hibernation, pigs undergo general anesthesia. CABG was performed via median sternotomy, and LIMA was dissected free from the chest wall, followed by the administration of heparin 200 units/kg. The left anterior descending (LAD) artery was then exposed distal to the stenosis and was prepared for LIMA–LAD anastomosis. Using the off-pump technique and coronary shunt, the anastomosis was performed using 7-0 prolene sutures. Immediately following anastomosis, a vicryl patch embedded with mesenchymal stem cells was sutured on the hibernating myocardial region using a polypropylene suture. The sternotomy was then closed.

### 4.4. Cardiac MRI

Cardiac MRI (CMRI) was performed to confirm >80% LAD stenosis and assess global and regional contractile function as well as diastolic function. Two CMRIs were performed—the first was performed 12 weeks after the placement of a constrictor band to assess for hibernation physiology, and the second was performed 4 weeks after revascularization to assess for improvement in cardiac function. CMRI was performed on a 1.5—Tesla clinical scanner, with the animals under general anesthesia. Following the acquisition of localizers, breath-held retrospectively triggered steady-state free precession cine images were collected in parallel short-axis planes from the base to the apex of the left ventricle to assess the global function and peak LV filling rate. This protocol was acquired during rest and repeated during dobutamine challenge using 5 μg/kg/min to simulate stress. Delayed enhancement images were acquired 10 min after injecting the gadolinium- based contrast agent gadobenate dimeglumine (Bracco Diagnostics Inc, Milan, Italy) to rule out infarction of the LAD territory. Remote segments of the left ventricle (LV) are defined as the posterolateral segments of the LV wall opposite the LAD distribution, LV global function, diastolic filling, and strain measurements were analyzed using CV142 (Circle Cardiovascular Imaging Inc, Calgary, AB, Canada). To limit the risk of bias, the MRIs were interpreted by blinded specialists.

### 4.5. Terminal Procedure

Four weeks after revascularization, a second cardiac MRI was performed, followed by sacrifice. Refer to Aggarwal et al. for detailed methods [9]. Briefly, pigs were anesthetized using general endotracheal anesthesia. Again, redo sternotomy was performed. Adhesions were dissected around the LIMA–LAD graft with extreme caution. The heart was then explanted. Coronary dilators were used to assess the degree of LAD stenosis and measure the vessel diameter proximal and distal to the bypass graft. All animals had a patent graft measuring 2.5–3.5 mm. Tissue was obtained from anteroseptal (ischemic region of LAD territory) distal to the stenosis and graft anastomosis as well as from the lateral left ventricular wall (remote region) to assess for the differences in inflammation.

### 4.6. Isolation of Bone Marrow-Derived Mesenchymal Stem Cells (MSCs)

We isolated and validated bone marrow-derived MSCs using a well-described method by Hocum Stone and colleagues [16]. Briefly, sterile bone marrow was collected from either the sternum or tibia of the swine in a BD vacutainer cell preparation tube, followed by centrifugation for 30 min at 1800× *g* to create a density gradient. The buffy coat with mononuclear cells was removed and washed with Hank’s balanced salt solution, followed by centrifugation to form a pellet of mononuclear cells, which was then resuspended in growth media (advanced Dulbecco’s modified eagle’s medium with 10% fetal bovine serum). Mononuclear cells were then transferred to cell culture flasks, and MSCs were isolated from the mononuclear fraction by their adherent nature. The phenotype of MSC was confirmed using flow cytometry.

### 4.7. Creation of an Allogeneic MSC Patch

Refer to Hocum Stone and colleagues for detailed methodology for creating an MSC-seeded patch [16]. Briefly, 4 × 10^6^ MSCs were seeded on a sterile polyglactin vicryl patch (approximately 3.5 cm diameter) and were incubated for 10 days until confluent. The patch embedded with MSCs was kept sterile and washed with sterile phosphate buffered saline (PBS), and it was then sutured onto the epicardium of the anterior portion of the heart, which is the territory of LAD distribution. Our patches held 8 × 10^6^ MSCs.

### 4.8. Histology

Myocardial samples from ischemic (anteroseptal) and non-ischemic (remote zone) regions were immediately rinsed and fixed in neutral buffered formalin before being embedded in paraffin and stained with hematoxylin and eosin (H&E), trichrome, and alpha-smooth muscle actin (SMA) staining for the analysis of tissue structure, viability, and collagen deposits, respectively.

### 4.9. Histologic Fibrosis and Inflammation Scoring

Refer to Aggarwal et al. for detailed methods on the scoring system [9]. Briefly, a scoring system was developed by an experienced animal pathologist to quantify fibrosis and inflammation among the study groups. Fibrosis grading was based on the location of fibrosis, and inflammation scoring was based on the number of inflammatory cells seen per high-power field. Measurements of areas were made using a calibrated combined microscope–camera–software system (Nikon Eclipse Ci microscope, Nikon digital Sight D5-U3 camera, NIS Elements D software version 4.5). To limit the risk of bias, all histological analyses were performed by a blinded animal pathologist.

### 4.10. Tissue and Protein Preparation

Refer to Aggarwal et al. for detailed methods on tissue and protein preparation and quantification [9]. Briefly, myocardium was harvested, flash-frozen in liquid nitrogen, and stored at −80 °C until used for assays. Before protein lysate preparation, the tissue was homogenized mechanically and using an automated homogenizer (gentleMACS Octo Dissociator; Miltenyi Biotec, Auburn, CA, USA). Aliquots of 50 mg tissue samples were then homogenized in 1x RIPA lysis buffer (Thermo Fisher Scientific, Waltham, MA, USA) with 2x HALT protease inhibitor cocktail (Thermo Fisher Scientific). Protein quantification was performed using a near-infrared spectrometer (Direct Detect; EMD Millipore, Burlington, MA, USA).

### 4.11. Western Blot

Western blot was used to identify and quantify the proteins PGC1α and NFκB. For detailed methods, refer to Hocum-Stone et al. [16]. Briefly, nuclear fractions were run in denatured and reduced conditions, and gels were loaded with equal amounts of protein. Proteins were separated by sodium dodecyl sulfate–polyacrylamide gel electrophoresis on a 10% tris-glycine gel and transferred to a nitrocellulose membrane using the semidry TransBlot Turbo system (Bio-Rad Laboratories, Hercules, CA, USA). Membranes were then blocked, probed with primary antibodies, and incubated with secondary antibodies (LiCor, Lincoln, NE, USA) for detection with the LiCor Odyssey Imager. The band density was determined using Image Studio 5.× software (LiCor) and normalized to total protein stain (REVERT total protein stain; LiCor).

### 4.12. Cytokine and Chemokine Analysis

The Milliplex MAP Porcine Cytokine/Chemokine Panel (PCYTMG-23K-13PX; Merck Millipore; Darmstadt, Germany) was used to measure GM-CSF, IFNγ, IL-1α, IL-1ra, IL-1β, IL-2, IL-4, IL-6, IL-8, IL-10, IL-12, IL-18, and TNFα. Plates were read using a Luminex Magpix (Luminex Corp; Austin, TX, USA). All samples were normalized using total protein.

### 4.13. Statistics

Statistical differences between groups were determined by using a one-way analysis of variance (ANOVA) followed by Tukey’s post hoc test, as we were interested in computing confidence intervals for every comparison among our four groups. The significance of cardiac function changes during rest and under stress were calculated using paired *t* tests. Statistics were calculated using GraphPad PRISM 9.0 software (GraphPad Software, Inc., La Jolla, CA, USA). A *p* value of <0.05 was used as the reference for statistical significance. Data are presented as the means ± SD for Figure 1 and as the means ± SEM for Figure 3 and Figure 4.

## 5. Conclusions

In our well-established, large animal model of HIB, we observed increased fibrosis and inflammation alongside decreased PGC1α expression and showed an incomplete recovery of global diastolic relaxation with CABG alone. The application of an MSC-loaded patch at the time of surgical revascularization significantly improved diastolic function during low-dose dobutamine infusion. Furthermore, increased PGC1α expression, as well as decreased inflammatory signaling and fibrosis, were observed following treatment with MSCs. These findings suggest that adjuvant cell-based therapy during CABG improves diastolic function by reversing hibernation-associated metabolic adaptations and downregulating profibrotic oxidant stress-inflammatory signaling in chronically ischemic myocardium. Future work will need to further elucidate the exact molecular mechanisms responsible for the stem cell-driven changes observed in this study and pave the way for stem cell patches to be used as adjunctive treatments for patients with ischemic heart disease.

## Figures and Tables

**Figure 1 ijms-24-05475-f001:**
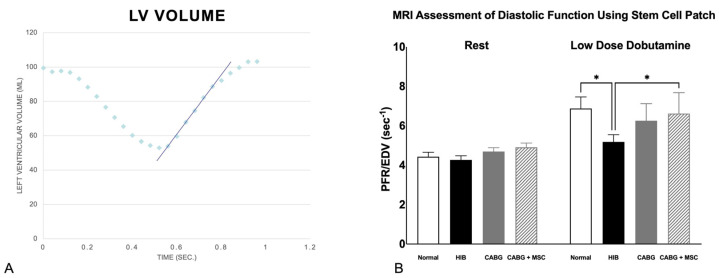
(**A**) Diastolic relaxation. Relationship of left ventricular volume throughout the cardiac cycle. X-axis: time in seconds; Y-axis: left ventricular volume in milliliters. The blue slope line represents the peak diastolic filling rate, noted as the fastest rate at which the left ventricle increases in volume. The highest positive slope, usually occurring just after the end-systole, represents the peak filling rate (PFR), which is normalized to the animal’s end-diastolic volume (PFR/EDV) to account for size variance across animals. (**B**) Measurements of Peak Filling Rates using cardiac MRI. Diastolic function, as measured by PFR/EDV, was compared among four groups (Control, HIB, CABG, and CABG + MSC). At rest, the PFR/EDV measurements are similar among the control (N = 6), Hibernating myocardium (HIB) (N = 17), CABG (N = 19), and CABG + MSC (N = 9) animal groups. However, significant differences exist during low–dose dobutamine infusion. Under stress, HIB animals show a significant decrease in PFR/EDV when compared to the control (* *p* < 0.05), with some improvement in the CABG-only group (*p* = 0.06) and a significant increase in the CABG + MSC group (* *p* < 0.05). A one-way analysis of variance (ANOVA) test was used for statistical analyses. Data are presented as the means ± SD. CABG = Coronary artery bypass graft, MRI = Magnetic resonance imaging, MSC = Mesenchymal Stem Cells.

**Figure 2 ijms-24-05475-f002:**
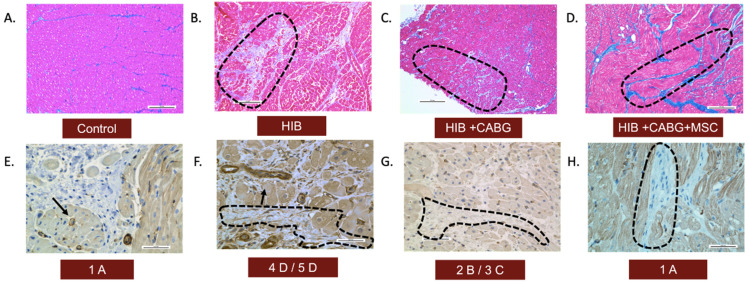
Histopathological assessment of inflammation and fibrosis. LAD tissue was obtained from animals in all four groups and was qualitatively analyzed for the presence of fibrosis and myofibroblasts. Trichrome stains (10×) of the LAD region showed (**A**) a normal amount of interstitial fibrosis in controls, (**B**) an extensive presence of fibrosis in endomysium in HIB animals, (**C**) the focal extension of fibrosis in endomysium in CABG animals, and (**D**) a normal amount of fibrosis, similar to the control, in the CABG + MSC group. The presence of myofibroblasts was assessed using alpha-SMA stain (40×). (**E**) In the control group, there was an absence of alpha-SMA stain in the myocardial tissue beyond the expected smooth muscle in the blood vessels (showed by arrows). (**F**) The HIB group showed an increased number of myofibroblasts between bundles of myofibers (encircled by black dashed line). (**G**) The CABG group showed very few myofibroblasts associated with alpha-SMA stain (encircled by a black dashed line), and (**H**) the CABG + MSC group showed minimal to no myofibroblasts. Scale bars are 0.2 mm. CABG = Coronary artery bypass graft, HIB = Hibernating myocardium, MSC = Mesenchymal Stem Cells.

**Figure 3 ijms-24-05475-f003:**
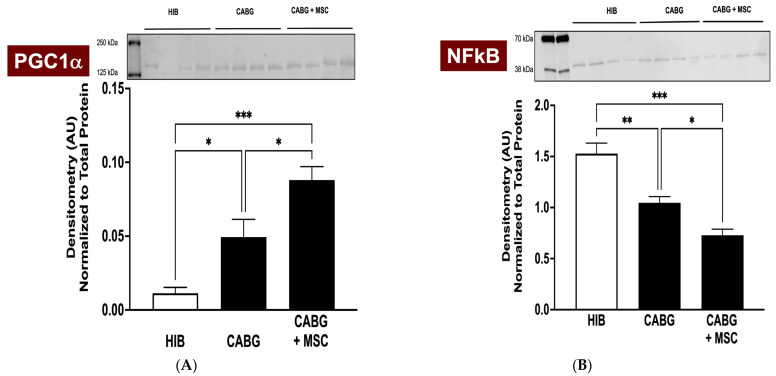
Western Blot analyses of PGC1α and NFκB. Porcine LAD tissue was obtained from three experimental groups (HIB, CABG, and CABG + MSC) to analyze the protein signal. (**A**) Western blots of CABG + MSC myocardial tissue show a dramatic increase in PGC1α compared to the HIB (*** *p* < 0.001) and CABG-only groups (* *p* < 0.05). (**B**) The expression of NFκB was measured in three experimental groups and was found to have a statistically significant decrease in CABG + MSC compared to the HIB (*** *p* < 0.001) and CABG groups (* *p* < 0.05). All protein level measurements were normalized to a total protein stain to correct for loading variability. One-way analysis of variance (ANOVA) and Tukey’s multiple comparison test were used for statistical analyses. Data are presented as the means ± SEM. * *p* < 0.05; ** *p* < 0.01, *** *p* < 0.001. CABG = Coronary artery bypass graft, HIB = Hibernating myocardium, MSC = Mesenchymal Stem Cells, NFκB = Nuclear Factor Kappa-Light-Chain Enhancer of Activated B Cells, PGC1α = Peroxisome Proliferator-Activator Receptor Gamma Coactivator 1-Alpha.

**Figure 4 ijms-24-05475-f004:**
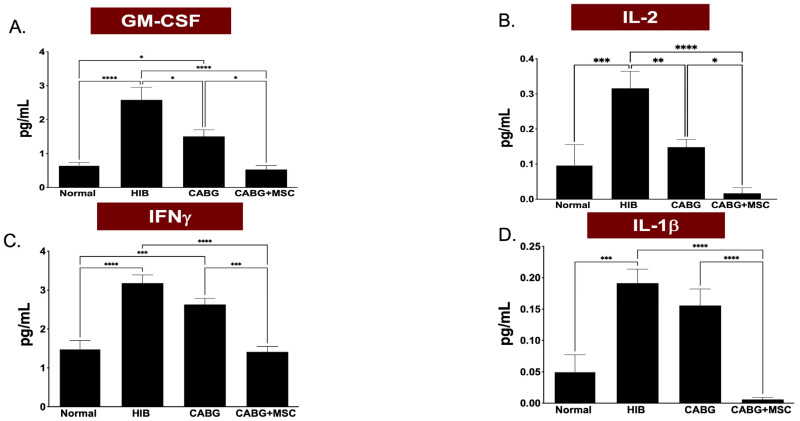
A multiplex cytokine and chemokine array analyses of GM-CSF, IFNγ, IL-2, and IL-1β isolated from porcine LAD tissue regions in the Control, HIB, CABG, and CABG + MSC groups (n = 5/group). (**A**) GM-CSF expression showed a significant increase in HIB compared to the control group (**** *p* < 0.0001), with a significant decrease in CABG + MSC when compared to the HIB group (**** *p* < 0.0001). (**B**) IL-2 expression showed a significant increase in HIB compared to the control group (*** *p* < 0.001), with a significant decrease in CABG + MSC when compared to the HIB group (**** *p* < 0.0001). (**C**) IFNγ expression showed a significant increase in HIB compared to the control group (**** *p* < 0.0001), with a significant decrease in CABG + MSC when compared to the HIB group (**** *p* < 0.0001). (**D**) IL-1β expression showed a significant increase in HIB compared to the control group (*** *p* < 0.001), with a significant decrease in CABG + MSC when compared to the HIB group (**** *p* < 0.0001). One-way analysis of variance (ANOVA), followed by Tukey’s post hoc test, was used for analyses. Data are presented as the means ± SEM. * *p* < 0.05; ** *p* < 0.01, *** *p* < 0.001, **** *p* < 0.0001. CABG = Coronary artery bypass graft, GM-CSF = Granulocyte-Macrophage Colony-Stimulating Factor, HIB = Hibernating myocardium, IFNγ = Interferon gamma, IL-2 = Interleukin 2, IL-1β = Interleukin 1 beta, MSC = Mesenchymal Stem Cells.

**Figure 5 ijms-24-05475-f005:**
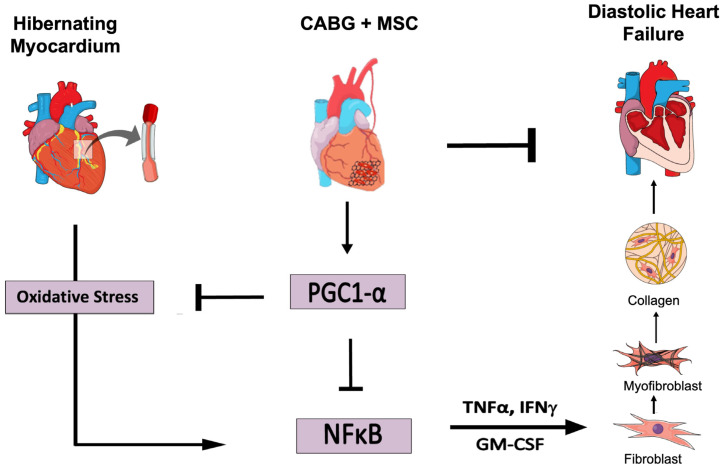
Adjuvant therapy with an epicardial placement of a mesenchymal stem cell (MSC) patch at the time of CABG improves diastolic function by reducing oxidative stress, pro-inflammatory signaling, and myofibroblast presence in myocardial tissue. CABG = Coronary artery bypass graft, GM-CSF = Granulocyte-Macrophage Colony-Stimulating Factor, IFNγ = Interferon gamma, IL-1β = Interleukin 1 beta, MSC = Mesenchymal Stem Cells, NFκB = Nuclear Factor Kappa-Light-Chain Enhancer of Activated B Cells, PGC1α = Peroxisome Proliferator-Activator Receptor Gamma Coactivator 1-Alpha, TNFα = Tumor Necrosis Factor. (Figure created in the Mind the Graph platform: www.mindthegraph.com, accessed on 25 August 2022).

**Table 1 ijms-24-05475-t001:** Qualitative assessment of fibrosis and inflammation in a swine model of hibernating myocardium using a scoring or grading system [20,21].

Grades of Fibrosis	Interstitial Fibrosis	Scoring of Inflammation	Severity of Inflammation
1	Limited in perimysium	A	No cells
2	Focally extending into endomysium	B	1–5 cells/HPF
3	Multiple extension into endomysium	C	5–10 cells/HPF
4	Diffusely extending into endomysium	D	11–20 cells/HPF
5	Replacement fibrosis	E	>20 cells/HPF

HPF = High-Power Field.

## Data Availability

All data are available within this article.

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
