# Peer review of "An Adjuvant Stem Cell Patch with Coronary Artery Bypass Graft Surgery Improves Diastolic Recovery in Porcine Hibernating Myocardium"

_ijms, 2023, doi:10.3390/ijms24065475_

Round 1
Reviewer 1 Report
I read with great interest the article entitled « Adjuvant Stem Cell Patch with Coronary Artery Bypass Graft Surgery Improves Diastolic Recovery», that concerns an important and clinically interesting issue. I have few comments and suggestions regarding this manuscript.
Major:
The title of the article states "...Stem Cell Patch with Coronary Artery Bypass Graft Surgery Improves Diastolic Recovery", that is, the primary study end-point was the changes in LV diastolic function/dysfunction. However, LV diastolic function was assessed using only one variable – MRI-derived PFR/EDV ratio, that limits the significance of the article. An increase in PFR (an analogue of echo E velocity or time velocity integral of transmitral E flow) indicates an improvement of LV diastolic function only at initially isolated impaired relaxation; however, in case of initially increased LV filling pressure (that may occur in severe myocardial hibernation) an increase in PFR is rather a sign of further LV filling pressure increases, for example, due to reperfusion injury.
Minor:
1. Line 79: Please, specify Ð level. Were the groups comparable by LV EF increase during dobutamine (i.e., systolic reserve)? It seems that the groups differed significantly in it (from 10 to 20 absolute %)
2. Line 87: Since the importance of normal systole for normal diastole, I suggest to make a correlation analysis between contractile and diastolic reserves in groups. It also is reasonable to discuss it.
3. Line 91: Please discuss why low dose dobutamine infusion was used to assess LV diastolic reserve and not, for example, treadmill exercise, that imposes a much greater load on the LV diastolic reserve than dobutamine test
4. Line 92: Please do not duplicate in the text the numbers that are presented in figures
5. Line 96: There was no P level for the difference between the CABG + MSC vs. CABG alone by PFR/EDV ratio. The lack of such difference casts doubt on the assumption that "…adjuvant cell-based therapy during CABG recovers diastolic function..." behind conventional CABG
6. Line 100: I think it is appropriate to present in Figure 1 for clarity an additional figure illustrating the example/calculation of PFR/EDV ratio.
7. Line 100: I think it is appropriate to present also as diastolic reserve – degree of increase in PFR/EDV ratio during dobutamine test
8. Line 115: Please discuss the presence of replacement fibrosis in the HIB group
9. Line 137: Please provide a link
10. Line 142: Please decipher the abbreviation PGC1α
11. Line 144: Please provide a link
12. Line 336: Please, specify “…tissue viability…”
13. Line 449: Please give the full link
Reviewer 2 Report
In this paper, Rishav Aggarwal et al highlight the potential of MSC patch application as adjunctive therapy at the time of CABG, to improve PGC1α expression and diastolic function along with decreased inflammatory signaling and fibrosis.
There are some major and minor comments to address.
Major comments
1) Abstract is missing background, methods, results, conclusions section. You should put the information in this order.
2) The introduction section must be mor focus on the essential background for the study. In some parts it looks like a discussion section or a review.
3) In the result section, about the increase in EF with dobutamine, you need to add also the delta [LVEF(dobutamine)-LVEF(rest)]. Moreover, you should clarify before what dose of dobutamine you used and considered low-dose vs. high-dose.
4) Why did you put references in the result section? In this section you should show only your results.
5) You spoke about “regional function” at MRI expressed in percentage. You should clarify better what parameter are you talking about.
6) Caption of figure 4 should be move to the results section.
7) The discussion section should be rewritten taking into account that the reader has to recognize the important findings and the explanation/revision of literature
Minor comments
1) Space between words.
2) Some abbreviation (like EDV) should be written in full for the first time.
3) English spelling
4) Figure 5 is good. If you use any software or images from other journals ore sources, you should have a license or you should declare in the caption.
Reviewer 3 Report
Dear Authors,
you aim to investigate to effect of a stem cell patch on hibernating heart after revascularization. This is an interesting and important scientific issue. you use a porcine model with LAD obstruction and operative revascularization for that.
There are some question, which should be solved.
Results
You talk about hibernating myocardium (HIB). In lines 79ff you state a “preserved ejection fraction among all four groups”. i don’t think the numbers are significantly different. however, there is not p value given. HIB is defined by reduced systolic function at resting conditions, which improve upon revascularization. the described model in this study is different. please clarify.
Further, figure 1 shows no significant difference between cabg + stem cell therapy and its respective negative control (cabg – stem cell). thus, your conclusion/title is not supported. there might be differences in protein expression, inflammatory response; however, it looks like these do not translate into reduced systolic function, nor into improved diastolic function with the use of revascularization/stem cells.
Method section
Please explain why you chose operative myocardial revascularization instead of removing the obstruction itself or pci +- stem cells?
Two MRIs were performed. please state that more clearly in the design section. in the MRI section, you should give data on the MRI method itself: which scanner, how much contrast medium, how was infarction ruled out (if you mean LGE positivity, please define it).
please define regional function.
Reviewer 4 Report
ijms-2190592, Adjuvant Stem Cell Patch with Coronary Artery Bypass Graft Surgery Improves Diastolic Recovery by Rishav Aggarwal et al. The authors aimed in their study to investigate whether the adjunctive use of mesenchymal stem cells patch during CABG surgery can improves diastolic function by reducing inflammation and fibrosis, in experimentally induced hibernating myocardium model.
Abstract:
- Page 1, lines 28-29: “These findings suggest that adjuvant cell-based therapy during CABG recovers diastolic function by reducing oxidant….”. The reviewer suggests easing the language of conclusion by changing “recovers” to “may recover”.
Introduction:
- Page 2, line 49: Reference #8 “Cabrera, J.A., et al., Uncoupling protein-2 expression and effects on mitochondrial membrane potential and oxidant stress in heart tissue”. The reviewer believes this reference may be irrelevant. Please check and edit accordingly.
- Page 2, line 72: “Adult-derived stem cells are promising ……. can expand ex vivo”. Can the authors give some examples and cite the appropriate references.
- Page 2, line 73: “Considering the potential of MSCs to enhance PGC1α signaling”. Please add the appropriate reference(s).
- All abbreviations should be fully explained at their first mention in the manuscript. Examples: PGC1α, NF-κB, MSCs etc.
Results
- Please spell out abbreviation: “PFR/EDV (Page 2, line 89) and “LIMA” (Page 3, line 97).
- Page 5, line 161: “…to have statistically significant decrease…”. The reviewer suggests changing the sentence to “…to be significantly lower…”.
All figures and tables:
- All abbreviations should be listed in tables footnotes/figures legends in alphabetical order.
- Authors need to mention the used statistical analysis/test in every figure legend, where applicable.
- What the error bars on the figures refer to? SD or SEM? Please mention how the data were expressed in every figure legend. Example: “Data are presented as means ± SEM”.
- Repetitive values: Since the authors extensively mentioned the values of the measured parameters in the text, the reviewer suggests deleting the values from the figure legends and just mention the P value (more important for the reader) to avoid repetition/redundancy. Example: Figure 1: The PFR/EDV values at rest and under stress have been mentioned in detail in the manuscript text (page 3, lines 88-96). Please edit/delete accordingly. The same would apply to Figure 3 as well.
- Figures 1 and 3: Please mention what are the P values corresponding to the asterisk * ,**, and ***. For example: P<0.05, P<0.01, and P<0.001 may be used as equivalent to the use of *, **, ***, respectively.
Materials and Methods:
- The authors are strongly recommended to reveal more details (two or three sentences) about each and every one of the techniques/protocols used/applied in the current study, rather than just referring to prior literature.
- Page 10, “Terminal Procedure”: The authors mentioned “a second cardiac MRI was performed”. How many times cardiac MRI was performed for the studied groups and how long after the surgery? Please clarify under “Cardiac MRI” section on page 10.
- Page 10, lines 359-361: “Tissue was obtained from septal and ventricular wall regions distal to the stenosis and graft anastomosis as well as from the lateral left ventricular wall”. Could the authors clarify why samples were obtained from these specific regions?
- Page 10, line 370: Please spell out the abbreviation “PBS”.
Statistics:
- While the authors mention the t test and Mann-Whitney U test in the “Statistics” sections, it appears that these two tests do not actually exist in the current manuscript. The same would apply to non-parametric tests which apparently were not used as well. Please clarify and edit accordingly.
- Inversely, the authors, indeed, used one-way ANOVA; however, the test was not mentioned in the “Statistics” section although post hoc analysis were mentioned. Please edit accordingly.
- What was the basis the authors used for selection of post hoc analysis? Tukey’s or Holm-Šídák’s?
- Please indicate clearly in the “Statistics” section what level the statistical significance was set (P value)? 0.05 or something else?
- Please mention how the data were presented in the manuscript in general (means ± SEM or means ± SD)?
Discussion:
- Authors may consider discussing more about how the current findings could be beneficial/translated to human.
- Page 8, line 282: The reviewer suggests specifying a separate section titled “limitations of the study”. The lower number of groups especially the “CABG + MSCs” might be considered a limitation as well.
- Please spell out abbreviation: CAD (Page 6, line 203) and ETC (Page 7, line 259).
Round 2
Reviewer 2 Report
The authors addressed all the comments
Reviewer 3 Report
Dear authors,
Thank you very much for your answers.
The findings that reperfusion improves regional but not necessarily global LV function are not really presented in your text. Probably other readers will miss this data, since you talk about regional function and describe strain measurements.
Figure 1: Something is missing here. In the figure legend, the x-axis is described as time in seconds. The abbreviations are explained twice.
You responded to my last comment that "...we have proposed the use of mesenchymal stem cells as a complementary therapy at the time of CABG. When stem cells are used as a patch, we found a statistically significant improvement in diastolic function compared with hibernating animals." To prevent misunderstanding and better prepare the reader, I suggest adding "in porcine hibernating myocardium" to the title. Your mention of other experiments and molecular background in your point-by-point response makes it difficult to see the answer to the question. NFkB, etc. are not content of Figure 1, are they?
Thank you for better explanation of the model.
Thank you for explaining the MRI method.
Regional function: since the differences in your model occur in the specific area of coverage (by the LAD), you might consider adding physiological data to this region in addition to the global analysis, for which your model is not actually designed - as you explain in your first answer. Either explain and show the global function, or mention the regional function (which you actually do) and provide data on that as well. Otherwise one might get lost in reading/understanding the text.
Reviewer 4 Report
The reviewer would like to thank the authors for implementing the suggestions/corrections per first round of review. Below few MINOR comments and suggestions that might be further beneficial to the manuscript:
1) P-value reporting:
- Please report the exact value for P<0.03 (Page 3, line 102) and P<0.02 (Page 5, line 158).
- Page 5, line 158: The reviewer suggests reporting p value of p<0.0006 as p<0.001 to be consistent with the p value reporting in the manuscript.
- For consistency, please use the same format for P value either with space (P < ) or without space (P<), depending on the guidelines of the journal.
2) Abbreviations: Some abbreviations were not addressed properly. Example: “NF-κB” was not spelled out at the first appearance in text (page 5, line 159). On the contrary, “LIMA” abbreviation was spelled out twice on Page 9, line 324 and line 338). Please edit/correct accordingly.
3) Figure 1:
- Could the authors clarify why they added a new paragraph under letter “A” titled “Diastolic relaxation. Relationship of left ventricular volume throughout the cardiac cycle”? also why there is referral for “A” and “B” panels, although the figure it self does not have 2 panels. Please clarify and edit accordingly.
- Please do NOT include “ANOVA” and “HIB” in the list of abbreviations as they were already spelled out in the figure legend.
4) Figures 2, 3, and 4: Please make sure to use the same format for referring the panels. For example, in Figure 2, the authors used UPPERCASE letters at the end of the text (A), whereas they used lowercase letters before the text (a) in Figure 3 . Please report consistently, per journal guidelines.
5) Statistical Analysis: Please make sure to use the same font style/size throughout this section.
6) The authors may consider putting “Study Limitations” and “Conclusions” sections after “Discussion” section in the revised manuscript.
Round 3
Reviewer 3 Report
Dear Authors,
the manuscript improved and my concerns have been addressed.